# Opening opportunities for $K_d$ determination and screening of MHC peptide complexes

Janine-Denise Kopicki [1,2], Ankur Saikia [3], Stephan Niebling[4], Christian Günther[4], Raghavendra Anjanappa[3], Maria Garcia-Alai[4], Sebastian Springer [3✉] & Charlotte Uetrecht [1,2,5✉]

An essential element of adaptive immunity is selective binding of peptide antigens by major histocompatibility complex (MHC) class I proteins and their presentation to cytotoxic T lymphocytes. Using native mass spectrometry, we analyze the binding of peptides to an empty disulfide-stabilized HLA-A*02:01 molecule and, due to its unique stability, we determine binding affinities of complexes loaded with truncated or charge-reduced peptides. We find that the two anchor positions can be stabilized independently, and we further analyze the contribution of additional amino acid positions to the binding strength. As a complement to computational prediction tools, our method estimates binding strength of even low-affinity peptides to MHC class I complexes quickly and efficiently. It has huge potential to eliminate binding affinity biases and thus accelerate drug discovery in infectious diseases, autoimmunity, vaccine design, and cancer immunotherapy.

[1] Centre for Structural Systems Biology (CSSB), Leibniz Institute of Virology (LIV), Notkestraße 85, 22607 Hamburg, Germany. [2] University of Siegen, Am Eichenhang 50, 57076 Siegen, Germany. [3] Life Sciences and Chemistry School of Science, Jacobs University Bremen, Campus Ring 1, 28759 Bremen, Germany. [4] Centre for Structural Systems Biology (CSSB), European Molecular Biology Laboratory – Hamburg Outstation, Notkestraße 85, Hamburg 22607, Germany. [5] European XFEL GmbH, Holzkoppel 4, 22869 Schenefeld, Germany. ✉email: s.springer@jacobs-university.de; charlotte.uetrecht@cssb-hamburg.de

Major histocompatibility complex (MHC) class I molecules present the intracellular peptidome to cytotoxic T lymphocytes (CTL), which detect non-self-peptide/MHC class I (pMHC) combinations and induce apoptosis in the presenting cell. The identification of high-affinity virus- or tumor-specific peptide epitopes is key to novel immunotherapy approaches (in which antiviral or antitumor CTL are identified, stimulated, and reintroduced into the patient) and to developing efficacious peptide vaccines (as prioritized by the WHO[1]). This is currently done either by isolating pMHCs from patient samples, eluting the peptides, and sequencing them by mass spectrometry (MS)[2]; or by sequencing the virus or the exome of a tumor and predicting MHC-binding peptides[3].

The structural basis of MHC-peptide binding is well understood[4], with >1500 human pMHC structures in the protein data bank (PDB[5]). The peptide binds into a groove that consists of a β sheet topped by two parallel α helices, with its amino terminus contacting the A pocket at one end of the groove[6] and the C-terminal carboxylate held by a network of hydrogen bonds at the other end. These interactions limit the length of a high-affinity peptide to 8–10 amino acids[7]. In addition, side chains of the peptide (usually two) bind into other pockets at the bottom of the groove; one of them is always the F pocket (usually binding the hydrophobic side chain of the C-terminal amino acid). Together, these interactions define the peptide-binding motif of a particular class I allotype (such as xLxxxxxxV for HLA-A*02:01). About 16,000 class I allotypes have been described to date, with nine of them found in >75% of the Caucasian and Asian population[8].

Structural knowledge and extensive databases of eluted peptides have informed computational methods that predict tumor epitopes[9]. This approach, however, suffers from the uncertainty created by matching the eluted peptide to one of the 4–6 MHC class I allotypes present in a human being. Hence, to accelerate biological testing, it is desirable to rank candidate peptides by their binding affinity using a direct approach. Yet, to date, simple equilibrium binding assays that support high-throughput screening are not available because they require empty peptide-receptive class I molecules, which are conformationally unstable[7,10].

Recently, empty disulfide-stabilized class I molecules (dsMHC) have become available, possessing the same peptide- and T cell receptor-binding specificities and affinities as the wild type[11–13]. Due to their stability in the empty state and their rapid binding, these dsMHC lend themselves to high-throughput approaches. We have previously employed native mass spectrometry (nMS) to analyze pMHC complexes and shown that the dsMHC variant of HLA-A*02:01 (dsA2) can indeed be freed from the dipeptide used to facilitate refolding. The empty binding groove can then bind an exogenous peptide, with the resulting pMHC appearing as an additional signal in the mass spectrum[14]. nMS simultaneously detects all different mass species present in solution, while quaternary structures and non-covalent bonds can be preserved[15]. Even though in the gas phase, there is no longer an equilibrium of protein-ligand binding, previous studies have demonstrated that nMS effectively determines dissociation constants, as the method captures and reflects the equilibrium state of the solution[16–22]. Moreover, a small ligand has only minor influence on the ionization efficiency of a much larger protein complex. For example, Garcia-Alai & Heidemann et al. have determined the dissociation constants for clathrin-associated adapter protein-phospholipid complexes[23]. The work of Jecklin et al. also demonstrates how nMS can be used to quantitatively assess binding affinity using multiple protein-ligand systems (hen egg-white lysozyme and N,N′,N″-triacetylchitotriose; adenylate kinase and adenosine-5′-diphosphate; lymphocyte-specific kinase

and a respective inhibitor)[21]. Fundamental in all examples is that the peak area of the individual mass species is integrated and then the concentrations of protein, ligand, and the resulting complex are calculated.

Here, the use of nMS to assess the binding affinity of peptides to dsMHC is demonstrated and verified by nanoscale differential scanning fluorimetry (nDSF). nMS reveals the effects of truncations and modifications on the peptide binding affinity, binding of two truncated peptides to the same class I molecule, and the absence of cooperativity between the A and F pockets in peptide binding. The method is suitable for high-throughput applications and can be utilized to systematically investigate MHC class I-peptide binding.

## Results

**Peak intensity in nMS reflects peptide-MHC-binding affinity.** Empty dsA2 consists of the disulfide-stabilized heavy chain (HLA-A*02:01(Y84C/A139C)) and the light chain, beta-2 microglobulin (β₂m). Bacterially expressed dsA2 heavy chain and β₂m are folded in vitro into the dsA2 complex with dipeptides (GM or GL) and purified by size-exclusion chromatography[10,14]. During size-exclusion chromatography, the dipeptide is removed and was no longer detected by nMS, resulting in an empty binding groove (Fig. 1a, b). At a low acceleration voltage of 25 V in the collision cell, raw and deconvoluted spectra (Fig. 1c, d) demonstrate a stable complex of heavy chain and β₂m in the absence of any peptide. Some minor in-source dissociation (ISD, <5%) occurs—independently of either the addition of a peptide or the characteristics of the peptide in question—upon activation in the mass spectrometer, with heavy chain and β₂m detected individually. The remaining fraction of dsA2 carries a small molecule adduct (337 Da by tandem MS analysis) that is easily released at ≥50 V by collision-induced dissociation. By small-molecule MS, this was most likely identified as erucamide ((Z)-docos-13-enamide), probably originating from laboratory plastic ware as reported[24–26] (Fig. S1). This contaminant appeared consistently in all protein batches studied.

It is next examined whether nMS can differentiate the binding of high-affinity and low-affinity peptides by comparing the A2 epitope NV9 from human cytomegalovirus pp65 (sequence NLVPMVATV; theoretical dissociation constant, $K_{d,th} = 26$ nM predicted by NetMHC[9]) with the irrelevant YF9 (YPNVNIHNF; $K_{d,th} = 27$ μM) and with GV9 (GLGGGGGGV; $K_{d,th} = 2.7$ μM), a simplified NV9 derivative that retains the anchor residues, L and V. After a short incubation, dsA2 with 5× excess peptide is subjected to nMS with increasing acceleration voltages. Since β₂m dissociates from the heavy chain above 75 V[14], and so only results for 10, 25, and 50 V are considered (Fig. 2; raw spectrum in Fig. S2). While at 10 and 25 V, the distribution of the different mass species is very similar, the ratios change significantly at 50 V. This is a frequent observation with the electrospray ionization process in nMS, where non-covalent hydrophilic bonds such as those between dsA2 and high-affinity peptides are retained[16,27–29], but hydrophobic interactions are weakened. By increasing the acceleration voltage, the dissociation of a protein-ligand complex usually does not occur gradually but spontaneously beyond a certain threshold, at which an energetic state is encountered that denatures the complex, here between 25 and 50 V. Therefore, measurements at 10 V are used to calculate the dissociation constants.

In the presence of low-affinity YF9, the empty dsA2 molecule (43,733 ± 4 Da) generates the largest signal (56 ± 3% at 10 V; Fig. 2a; reference spectra in Fig. 2b). Most of the remaining dsA2 carries only erucamide (44,071 ± 5 Da; 39 ± 2%). There is very little dsA2/YF9 complex (4 ± 2%), and because of the very low

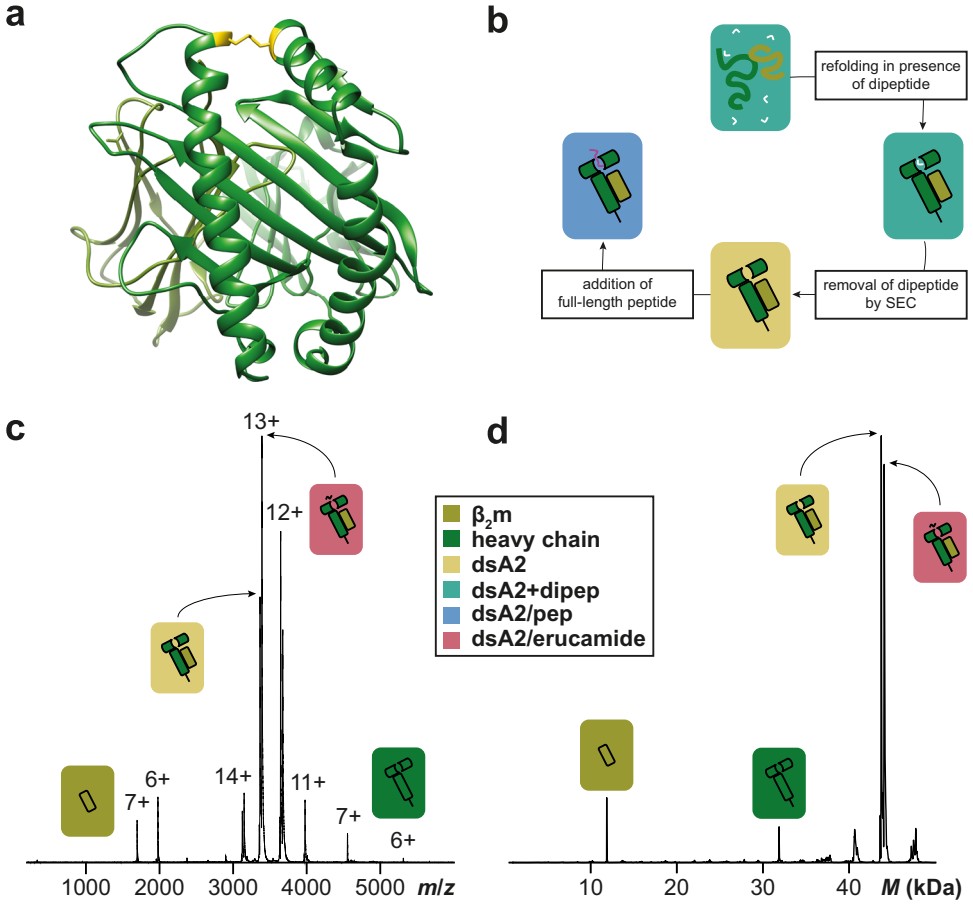

**Fig. 1 Disulfide-stabilized HLA-A*02:01. a** Crystal structure of empty dsA2 (PDB 6TDR[14,68]) seen as a top view of the peptide pocket. $\beta_2$m is shown in olive and the heavy chain in green. The stabilizing disulfide bond (positions 84 and 139 mutated to cysteines) is depicted in yellow. **b** dsA2 is refolded in presence of GM or GL dipeptide, which is then removed via size-exclusion chromatography. Afterward, full-length peptides can bind into the empty binding groove. Raw (**c**) and deconvoluted (**d**) native mass spectra of peptide-free dsA2 were recorded at an acceleration voltage of 25 V. The empty dsA2 is the predominant species (yellow). In addition, dsA2/erucamide (coral) and dissociated $\beta_2$m (olive) and heavy chain (green) can be seen.

binding affinity of YF9, it is assumed that this signal does not correspond to real binding but rather to an artifact of the electrospray process known as nonspecific clustering[17,30–33]. Assuming that the other tested peptides cluster to the same extent, all nMS data were corrected for the clustering determined with YF9, for which thus no affinity can be calculated (raw data see Table S2).

For NV9, in contrast, strong binding is observed, with 64 ± 3% for dsA2/NV9 at 10 V and 40 ± 4% at 50 V. The dsA2/erucamide complex is completely absent, suggesting that erucamide is displaced by NV9. Hence, erucamide either binds into the peptide groove, or binds elsewhere and is displaced by a conformational change caused by peptide binding. A small amount of another mass species (45,624 ± 4 Da) corresponds to dsA2 with two molecules of NV9 (dsA2/NV9/NV9) with an abundance of 4 ± 4% (10 V) and 1 ± 2% at 50 V is also observed. The latter is likely the result of unspecific clustering, as the abundance correlates with the intensity of the first binding event and is similar to the intensity of unspecific YF9 clustering. Within NV9, the leucine in the second position and the C-terminal valine bind into the B and F pocket of the binding groove, respectively[12,34,35]. The proportion of dsA2 occupied with GV9 at 10 V (43 ± 2%, and 1.5 ± 0.4% for dsA2/GV9/GV9) is significantly lower than the proportion of dsA2/NV9. Thus, the minimal binding motif cannot support the same affinity as NV9, suggesting that other amino acids contribute significantly to the binding. At 50 V, however, the abundance of the dsA2/pep complex is the same for

GV9 and NV9. This indicates that the strong B and F pocket side-chain interactions together with the binding of the termini determine pMHC gas-phase stability.

Despite strong binding with nM affinity[36], the obtained $K_d$ ($K_{d,high}$, defined below) for NV9 is only 8 ± 2 µM, and the expected fully occupied (>99%) peptide binding pocket is not observed (Fig. 2a). Protein denaturation due to storage or other stresses can be from standard control experiments. Instead, ISD is the cause. Indeed, lower, i.e., gentler, cone voltages[37,38] increase the occupancy of the peptide-binding groove with both the peptide and erucamide. This linear relationship is most evident with dsA2 and erucamide in absence of any peptide (Fig. 3a). The proportion of the dsA2/pep species is likewise linearly dependent on the cone voltage (Fig. 3b). The extent to which the protein-peptide complex is affected by ISD depends on the gas phase properties and the binding affinity of the peptide in question. The data suggest that at a cone voltage of ≤48 V (which is unfeasible experimentally), dsA2 is 100% occupied with erucamide. Thus, erucamide is used as a reference species for peptide binding measurements, assuming that in solution, all free dsA2 protein is initially bound to erucamide as suggested by the zero-cone voltage extrapolation, and that the peptide replaces it. This way, the fraction of peptide-free dsA2 (identical to the fraction of dsA2/erucamide complex) in the sample is calculated from a measurement at 150 V cone voltage using the correction factor of 2.2, and from this, the fraction of dsA2/peptide can be inferred (see Materials and Methods). The resulting $K_d$ is a reliable

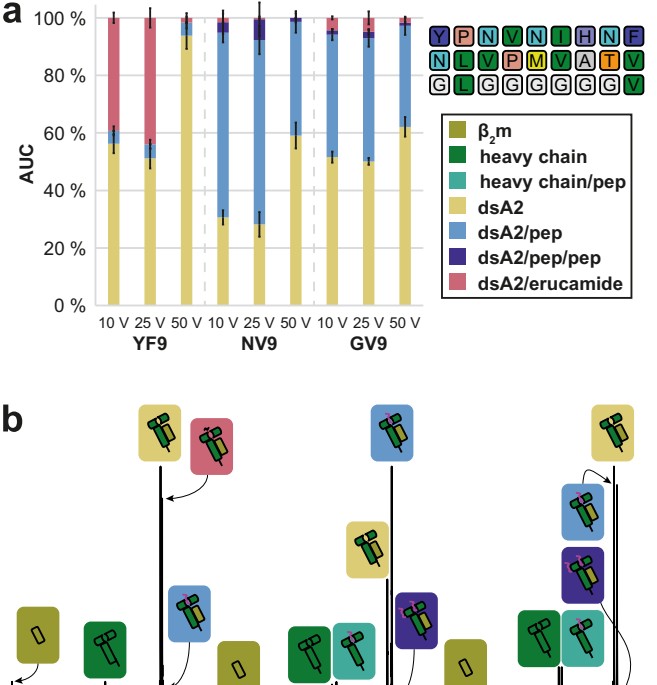

**Fig. 2 Overall area under the curve (AUC) for the detected dsA2 mass species in presence of YF9, NV9, and GV9 at different acceleration voltages. a** The AUC is determined over the entire spectrum for the respective mass species at 10, 25, and 50 V. The cone voltage is kept constant at 150 V. The mean value of the AUC in the absence or presence of the different peptides (protein–peptide ratio 1:5) from at least three independent measurements is depicted along with error bars that represent the corresponding standard deviation. "dsA2" corresponds to the empty HLA-A*02:01(Y84C/A139C) disulfide mutant complex, "dsA2/pep" to dsA2 bound to one peptide, "dsA2/pep/pep" to dsA2 bound to two molecules of this peptide, and "dsA2/erucamide" to dsA2 bound to erucamide, respectively. The negative control YF9 barely associates with dsA2, whereas the positive control shows a high proportion of dsA2/NV9, indicating high affinity. GV9, which contains only the two anchor residues Leu-2 and Val-9 of NV9, still shows a high affinity, and at 50 V, their dsA2/pep proportions are very similar, showing that their gas-phase stability is comparable. **b** Representative charge-deconvoluted spectra of the distinct protein and protein–peptide complex species recorded at 25 V. Olive and dark green correspond to the free β2m domain and heavy chain, respectively. Teal corresponds to a free heavy chain still attached to a peptide. The different complexes are assigned in yellow (empty dsA2), coral (dsA2/erucamide), light blue (dsA2/pep), and dark blue (dsA2/pep/pep).

approximation of the in-solution environment. Standardizing ISD to the dsA2/erucamide complex is advantageous since for individual pMHCs, the ISD is naturally influenced by peptide size and sequence, precluding compensation, while the ISD of the MHC-erucamide complex is invariant and peptide-independent. Therefore, for each peptide, both a dissociation constant for the high cone voltage (150 V) based on the clustering-corrected MHC-peptide signal ($K_{d,high}$) and another for the theoretical low cone voltage of 48 V based on the MHC-erucamide signal ($K_{d,low}$) are described (Table S3). While the experimentally determined $K_{d,high}$ is overestimated due to ISD, the $K_{d,low}$ is a more reliable approximation, well demonstrated by the example of NV9. In the following figures, therefore, $K_{d,low}$ is given.

Next, a series of sixteen peptides, most of them variations or truncations of the sequence of NV9, are compared in nMS and in

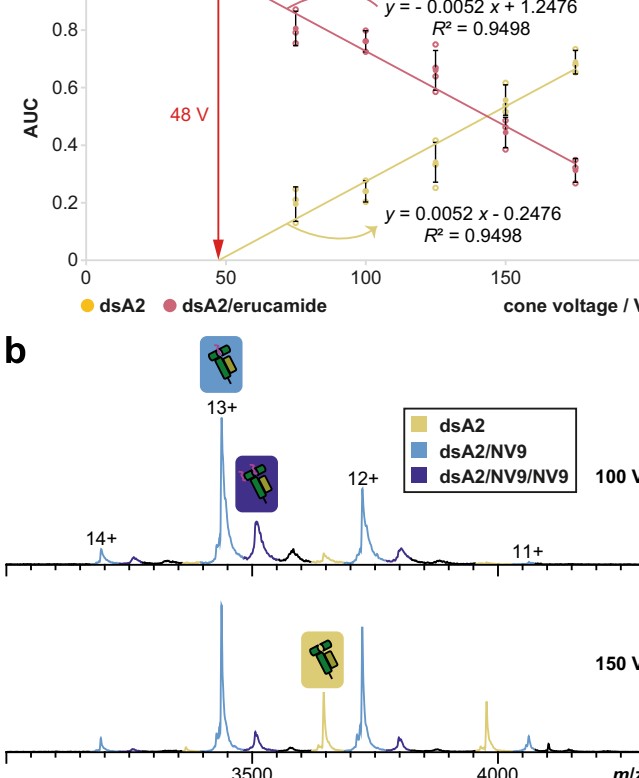

**Fig. 3 In-source dissociation causes linear dependence of the ligand-bound dsA2 fraction on cone voltage. a** Native MS fractions of dsA2 and dsA2/erucamide are likewise dependent on the activation energy of the cone voltage. As the cone voltage is raised, the percentage of empty dsA2 increases because the adduct binding cannot withstand the energy in the ion source and therefore dissociates. The proportionality is linear, and therefore a theoretical cone voltage of 48 V can be calculated at which a full occupancy with ligand would be reached. This voltage is too low to obtain stable electrospray and a resolved spectrum. **b** Peptide binding is equally subject to ISD. NV9-bound dsA2 (light blue and dark blue annotated peaks) become less prominent with increasing cone voltage, while empty dsA2 (yellow) increases at the same time.

the previously validated method, isothermal analysis of nanoscale differential scanning fluorimetry (iDSF)[39]. $K_d$ values from nMS and iDSF correlate very well over a wide range (Table S3 and Fig. 4a). It has to be noted though that affinities below 200 nM have to be treated with caution as the peptide concentration, and hence the fraction of dsA2/pep, is very low in iDSF.

It was also found that $K_d$ values from nMS correlate well with the thermal stabilization $\Delta T_m$ of dsA2 by peptide binding (protein-peptide ratio: 1:10, Fig. 4b), i.e., the increase in the midpoint of thermal denaturation ($T_m$) above that of the empty dsA2 ($35.7 \pm 0.6\,^{\circ}$C) as measured by tryptophan nanoscale differential scanning fluorimetry (nDSF[40,41]). While the negative control YF9 clearly shows no stabilization ($\Delta T_m = 1 \pm 1$ K), the positive control NV9 shows a high $\Delta T_m$ (23.4 ± 0.6 K) in agreement with published data;[12] this represents the maximum $\Delta T_m$ possible since NV9 stabilizes the binding groove so strongly that upon heating, other domains unfold first (Fig. S2)[40,41]. All other peptides show an excellent correlation between $K_d$ and $\Delta T_m$. Those identified as strong binders by their $K_d$ exhibit a $\Delta T_m$ of at least 6.4 ± 0.6 K (GV9), while the $\Delta T_m$ for the low-affinity peptides ranges between ≈0 and ≈3 K (Fig. 4b, red area). A closer evaluation of the data shows that peptides with a $K_{d,low} < 1\,\mu$M

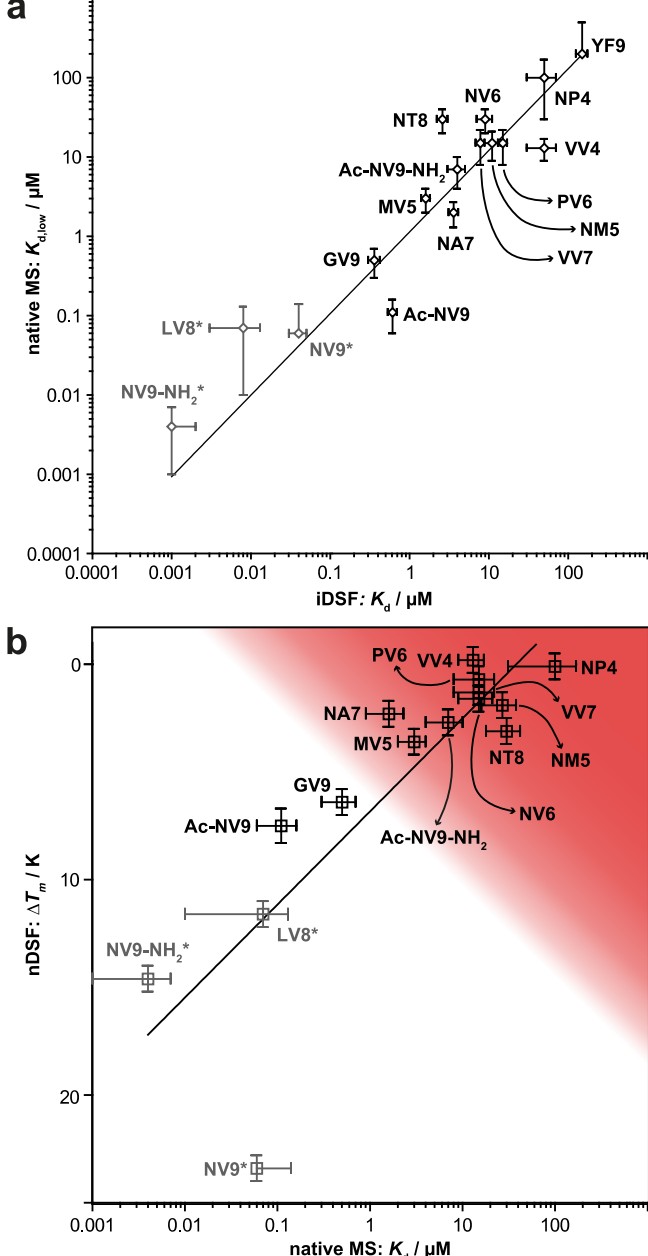

**Fig. 4 Relation of thermal stability and affinity for dsA2/peptide complexes. a** Affinities determined via iDSF and native MS correlate. Displayed data points represent the relationship of both apparent $K_d$ for all different dsA2/peptide systems analyzed. Both axes are scaled logarithmically. **b** Thermal denaturation measurements availing intrinsic tryptophans' change in fluorescence are used to define protein complex stabilization upon peptide binding, whereas an apparent $K_d$ for the various peptides is determined using native MS (10 V acceleration voltage and 150 V cone voltage). The dsA2 complex and peptide are deployed in a ratio of 1:10 ($\Delta T_m$) or 1:5 ($K_d$) depending on the experiment. Peptides showing a small $K_{d,low}$ for binding dsA2 and being concomitantly able to stabilize the protein complex are defined as strong binders (white area). The remaining peptides (red area) lack crucial features, making them unable to form strong bonds to dsA2 indicated by low binding affinities and melting temperatures. The standard deviation for both methods is displayed by error bars. The x coordinate is displayed logarithmically. *iDSF reaches its limits at affinities below 200 nm, hence the values of grayed-out peptides are not reliable.

also have an increased gas-phase stability of the complex at 50 V (dsA2/pep + dsA2/pep/pep > 35%) and present only negligible amounts of dsA2/erucamide at any voltage. Therefore, we define all peptides that fall within this range as strong binders for dsA2 (Fig. 4b, white area). These results demonstrate that by simply comparing the intensities in the native mass spectrum, it is possible to determine the $K_d$ and hence high-affinity epitopes for HLA-A*02:01.

**Neutralizing the terminal charges of the peptide reduces binding efficiency.** Next, the influence of the charged termini of the peptide upon the $K_d$ is analyzed. We hereby simulate the binding of deca- or undecapeptides but also peptides with common naturally occurring modifications in eukaryotes such as $N^\alpha$-acetylation[42,43]. For this purpose, three variants of NV9 are designed: Ac-NV9-NH2 has an acetylated N-terminus and an amidated C-terminus, whereas Ac-NV9 and NV9-NH2 each carry only one of these modifications. For the dsA2/pep fraction at 10 V, Ac-NV9 and NV9-NH2 show only a small difference from unmodified NV9, with comparable apparent $K_d$ values ($K_{d,low}$ = 0.11 ± 0.05 μM and 0.004 ± 0.003 μM). Further, no increase of dsA2/erucamide is observed (Fig. 5a; reference spectra in Fig. 5b). For both peptides, the protein-ligand complex is still stable at 50 V (Fig. S5). For Ac-NV9, the proportion of the occupancy is even higher than for NV9 itself (57 ± 2% and 8 ± 1% vs. 40 ± 4% and 1 ± 2%, Table S2). Remarkably, all modified peptides have an increased double occupancy. For the previously discussed peptides, dsA2/pep/pep is significantly lower with the result that correction for nonspecific clustering reduces it to a value below a threshold. For Ac-NV9, this effect is significant, since even at 50 V the proportion of dsA2/pep/pep (dark blue) is still 8 ± 1% indicating altered clustering or self-interaction. This clearly shows the benefit of the indirect erucamide approach. However, the stabilization effect on dsA2 in a 1:10 thermal denaturation approach is rather moderate for Ac-NV9 ($\Delta T_m = 7.5 ± 0.8$ K), while it is very strong for NV9-NH2 ($\Delta T_m = 14.6 ± 0.6$ K), indicating that the N-terminus has more relevance for tight peptide binding than the C-terminus.

For Ac-NV9-NH2, the affinity decreases significantly ($K_{d,low}$ = 7 ± 3 μM) due to the dual modification, and at an acceleration voltage of 50 V, the complex is no longer stable. The nDSF measurements likewise show that the stabilization effect with $\Delta T_m = 2.7 ± 0.6$ K is rather weak. At the same time, there is no significant increase in the formation of double binding events.

In summary, NV9-NH2 is far better rated in terms of affinity and stabilizing effect (Fig. 4a), but Ac-NV9 still falls into the category of a strong binder. For Ac-NV9-NH2, which no longer carries terminal charges, a loss of binding efficiency is observed, indicating that the respective hydrogen bonds are indispensable in the formation of the MHC-peptide complex.

**NV9 truncations disclose preferred binding positions in the peptide.** Since the anchor residues do not determine affinity alone, the effect of stepwise truncation of NV9 from either terminus on binding is analyzed. Building on the knowledge from the dsA2/NV9 crystal structure[12,14], the peptide-binding groove can thus be mapped and further understood. In vivo, MHC class I predominantly binds peptides with a length of eight to ten amino acids[7,44,45]. It is therefore not surprising that octa- and non-apeptides show the highest affinity (Fig. 6 and Table S3). Still, the effects of the loss of either terminal amino acid are strikingly different. Without the N-terminal asparagine (LV8), the affinity ($K_{d,low} = 0.07 ± 0.06$ μM) is still comparable to NV9, with considerable thermal stabilization (1:10, $\Delta T_m = 11.6 ± 0.6$ K). At 50 V, dsA2/LV8 even appears to be more stable than dsA2/NV9.

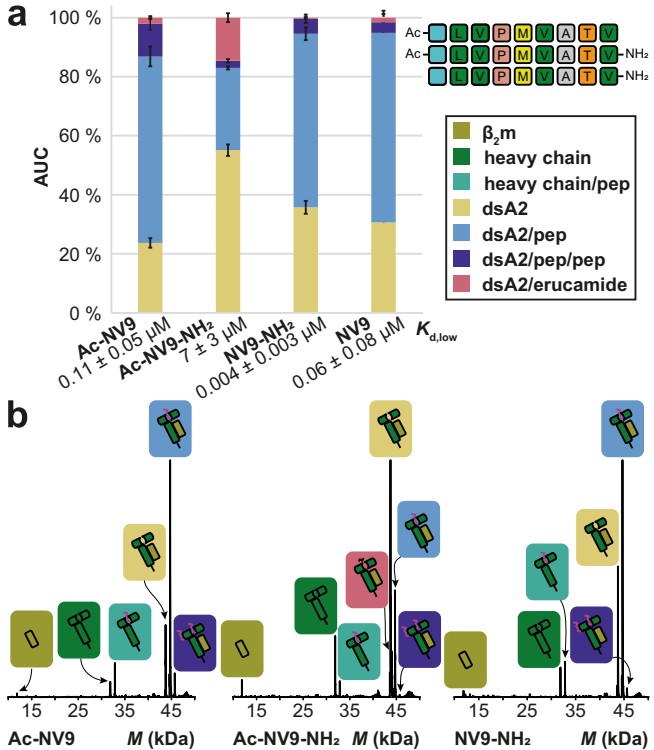

**Fig. 5 Binding of charge-reduced NV9 variants. a** The AUC is determined over the entire spectrum for the respective mass species, which is depicted along with error bars that represent the corresponding standard deviation. "dsA2" corresponds to the empty complex, "dsA2/pep" to dsA2 bound to one peptide, "dsA2/pep/pep" to dsA2 bound to two molecules of this certain peptide and "dsA2/erucamide" to dsA2 bound to erucamide respectively. By modifying only one terminus (Ac-NV9 and NV9-NH₂), the affinity of the peptide to dsA2 changes only marginally, but if the charges on both termini are neutralized (Ac-NV9-NH₂), the peptide binding is greatly reduced. The corresponding $K_{d,low}$ is shown for respective peptides. **b** Representative charge-deconvoluted spectra of the distinct protein and protein–peptide complex species recorded at 10 V. Olive and dark green correspond to the free β₂m domain and heavy chain respectively. Teal corresponds to a free heavy chain still attached to a peptide. The different complexes are assigned in yellow (empty dsA2), coral (dsA2/erucamide), light blue (dsA2/pep), and dark blue (dsA2/pep/pep).

In contrast, loss of the *C*-terminal valine (NT8) strongly decreases affinity ($K_{d,low} = 30 \pm 10$ μM). The erucamide is no longer fully displaced by NT8, only a small fraction of the protein–peptide complex is resistant to 50 V acceleration voltage, and no significant stabilization effect ($\Delta T_m = 3.1 \pm 0.6$ K) is measured. Thus, the interactions of the *C*-terminal amino acid, an anchor residue, are more important for overall binding than those of the *N*-terminal amino acid. However, when the second amino acid from the *N*-terminus, leucine, is also eliminated (VV7), affinity and stabilization are strongly diminished as expected, since the second position is an anchor residue. Strong binding between peptide and A2 only occurs if both B and F pockets are occupied with peptide side chains. Still, the difference between GV9 and LV8 points towards an additional contribution of the other amino acids that is in sum larger than the *N*-terminal contribution.

While none of the examined tetra- to heptapeptides are strong binders, VV7, still containing the *C*-terminal anchor residue Val-9, performs worse than NA7, which in contrast carries Leu-2. NA7 displaces more of the erucamide, and at 50 V, significantly more NA7 than VV7 is observed (Fig. S5). Thus, Leu-2 appears to impart stronger binding than Val-9. Curiously, the *N*-terminally

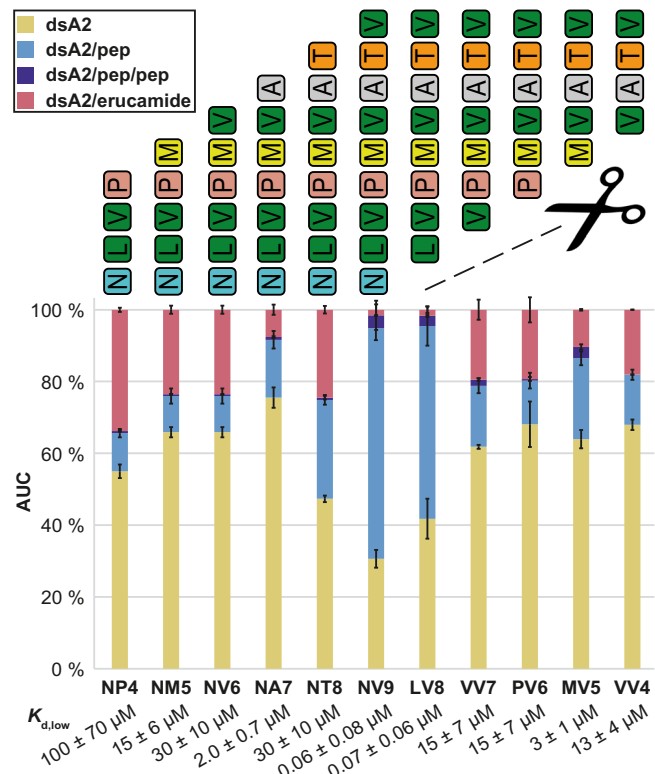

**Fig. 6 Truncated NV9 variants analyzed at 10 V.** The AUC is determined over the entire spectrum for the respective mass species, which is depicted along with error bars that represent the corresponding standard deviation. "dsA2" (yellow bars) corresponds to the empty complex, "dsA2/pep" (light blue bars) to dsA2 bound to one peptide, "dsA2/pep/pep" to dsA2 bound to two molecules of this certain peptide (dark blue bars) and "dsA2/erucamide" to dsA2 bound to the erucamide (coral) respectively. Octa- and nonapeptides have the highest binding efficiency in accordance with their biological function. Without the anchor residues Leu-2 and Val-9 in a truncated NV9 variant, the affinity decreases drastically. A terminal methionine appears to enhance the affinity of the peptide to dsA2. The corresponding $K_{d,low}$ is shown for respective peptides.

truncated pentapeptide MV5 performs significantly better than the corresponding hexa- (PV6) and heptapeptides (VV7). The increased proportion of dsA2/MV5 at 50 V (Fig. S5) and the simultaneous occurrence of dsA2/MV5/MV5 (dark blue bars, Fig. 6) strongly suggest that MV5 occupies an additional binding site within the peptide groove. Since this effect is not apparent for the smaller VV4, the terminal methionine seems to be decisive. Since no increased dual occupancy is observed for NM5, and since binding at 50 V is weaker than for MV5, it seems that the terminal methionine as such is not determinant on its own, but rather the relative position within the peptide. Considering the tetrapeptides, again the *N*-terminally truncated peptide, VV4, binds better than the *C*-terminally truncated NP4 at all acceleration voltages, pointing to the positioning of the erucamide within the peptide pocket. While NP4 carries only the anchor peptide Leu-2, it seems that VV4 has a higher chance to bind to the dsA2 peptide groove due to its two terminal valines.

**The binding groove may be occupied simultaneously by two short peptides.** Next, the simultaneous presence of two peptides in the binding pocket is further investigated[46]. Since nMS measures the binding of either peptide independently of the presence of the other in a single experiment, this allows us to assess whether peptide binding to the two ends of the peptide-binding

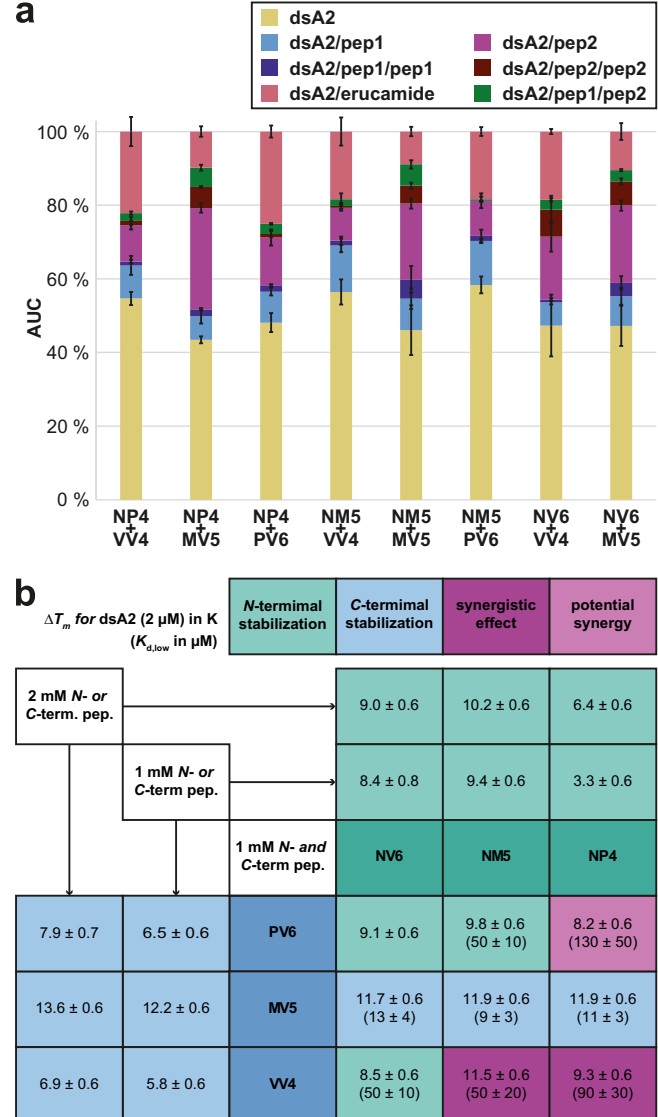

**a**

**b**

Fig. 7 Detected dsA2 mass species in presence of two corresponding truncated NV9 variants. a The native MS data suggest that the truncated peptides do not bind cooperatively as the amount of dsA2/pep1/pep2 remains small in all measurements. Rather, the affinity of the individual peptides is independent of each other. The AUC is determined over the entire spectrum for the respective mass species at 10 V. The mean value of the AUC in the absence or presence of the different peptides (protein–peptide ratio 1:10:10) from at least three independent measurements is depicted along with error bars that represent the corresponding standard deviation. "dsA2" (yellow bars) corresponds to the empty HLA-A*02:01(Y84C/A139C) disulfide mutant complex, "dsA2/pep" (light blue bars) to dsA2 bound to one peptide, "dsA2/pep/pep" to dsA2 bound to two molecules of this certain peptide (dark blue bars), "dsA2/pep2" to dsA2 bound to another peptide when two different peptides were present (purple bars), "dsA2/pep2/pep2" to dsA2 bound to two molecules of the second peptide (dark red bars), "dsA2/pep1/pep2" to dsA2 bound to one molecule of each of both peptides (dark green bars) and "dsA2/erucamide" to dsA2 bound to the erucamide (coral bars) respectively. The corresponding $K_{d,low}$ is shown for respective peptide pairs in (**b**). **b** The matrix visualizes the changes of $T_m$ in case half of the peptide amount is either exchanged for the corresponding N- (teal) or C-terminal (blue) peptide respectively or alternatively omitted. Teal marks the occurrence of major N-terminal stabilization whereas, for the light blue cells, the complex is primarily stabilized by the C-terminal peptide. Purple cells show peptide coupling with a significant synergistic effect or potential synergy (light purple).

NP4 with PV6. This once again suggests that VATV appears to have more binding options due to its two terminal valines.

In addition to the nMS measurements, nDSF measurements were performed with higher peptide excess to evaluate whether two peptides are able to synergistically stabilize dsA2. The $T_m$ of dsA2 exposed to one of the truncated NV9 variants (Fig. 3) is compared to the resulting $T_m$ with two peptides, while the ratio of protein to total peptide concentration is kept constant at 1:1000. If no synergistic effect occurs, then the resulting $T_m$ is expected less than or equal to that of the peptide complex with higher affinity. In line with the nMS results, most peptide pairs do not show additional stabilization by peptide combinations. Only for NP4 and VV4 (and to a lesser extent for NP4 and PV6), a higher $T_m$ is measured for both peptides together, which demonstrates that dsA2 can be stabilized by two shorter peptides in a synergistic fashion if they do not compete with each other for binding (Fig. 7b). Once again, MV5 stands out particularly in this experiment. Two parts of MV5 provide significantly more thermal stability to dsA2 than one part of MV5 together with one part of one of either of the corresponding N-terminal peptides, again suggesting that MV5 can bind into two positions in the binding groove.

Taken together, these observations suggest that the unfolding of MHC class I (or at least of the disulfide mutant) can begin at either end of the peptide-binding groove and that the opposite ends of the binding groove cannot communicate conformationally, which was already described for the corresponding wild type[11].

## Discussion
While only wild-type pMHC have been studied by native MS so far[47], we have recently demonstrated that peptides added to empty disulfide-stabilized class I molecules can be detected as well[14]. The present study shows that this method—being fast and amenable to high-throughput approaches—can be used to measure MHC class I-peptide binding affinities and to map the contributions of parts of the peptide to high-affinity binding.

groove is cooperative. The truncated peptides of NV9 are combined with their corresponding counterparts from the opposite terminus. Two short peptides that bind to different sites in the binding groove could either bind independently without any communication between the binding sites or in a cooperative manner, where the two binding sites communicate by a conformational or dynamic change in the protein and one peptide stabilizes (or destabilizes) the binding site for the other, such that the binding affinity for the second peptide is higher (or lower) in presence of the first.

Figure 7a shows single and double occupancy in nMS measurements for either peptide and for both together. The affinities of the two individual peptides found earlier are reflected well in this combined experiment. No double-occupied complex can be detected in the pairing of the pentapeptide NM5 and the hexapeptide PV6, presumably, due to spatial restriction. In the other combinations, dsA2/pep1/pep2 is detected. Yet only when MV5 is involved in the binding, the complex can withstand higher acceleration voltages. It is assumed, however, that this is not due to a cooperative binding between NP4 and MV5, NM5 and MV5 or NV6 and MV5, but can rather be explained by the higher affinity of MV5 described above. Remarkably, VV4 achieves a significantly higher occupancy when combined with the corresponding hexapeptide (NV6), which is not the case for

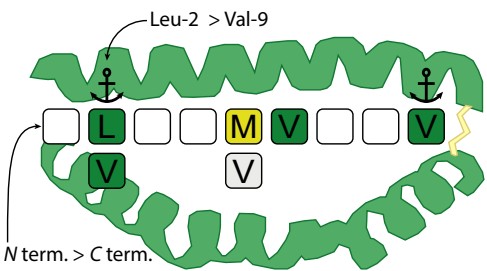

**Fig. 8 Favored amino acid positions within the HLA-A\*02:01 peptide-binding pocket.** Native MS confirms Leu-2 in the B pocket and Val-9 in the F pocket respectively as the main anchor positions of the pMHC. The analysis of truncated NV9 variants reveals that Val-2 is also a favored binding site. Moreover, Met-5 or Val-5, as well as Val-6, also contribute significantly to binding under certain circumstances. Among the anchor residues, Leu-2 contributes significantly more to the binding than Val-9. Concerning the termini, the *N*-terminus is of greater importance for binding than the *C*-terminus.

Our data affirms the key interactions between dsA2 and its high-affinity ligand NV9 that were described previously:[48–50] The termini as well as Leu-2 and the *C*-terminal Val-9, both binding into specificity pockets, contribute the major portion of the binding energy. For the F pocket, it is well established that Val or Leu are the strongest anchor residues with Val being the more preferred one, but our data suggest that MVATV can also stabilize the B pocket via its first valine similarly and possibly even simultaneously. The extraordinary thermal stabilization observed for dsA2/MV5, especially in comparison with potentially synergistic peptide pairs, gives rise to the assumption that even more than two binding modes are available for MV5. As reported before[51], along with leucine and methionine, valine is indeed one of the amino acid residues preferred by the A\*02:01 and moreover by the entire A2 supertype as an anchor residue in the second position. Since proline in the *N*-terminal position was found to be a deleterious factor for binding, this could explain the poor performance of PV6 in our experiments despite the preferred methionine in the second position. Surprisingly, NV6 scores low in all our tests despite having leucine in the second position and a *C*-terminal valine. Given the two different binding modes, at least a higher proportion of singly-bound species would be expected here—similar to MV5—for statistical reasons. A potential explanation could be self-competition of the peptide for the two separate binding sites. Based on their sheer size, two NV6 molecules sterically hinder each other as opposed to the pentapeptide MV5. As previously demonstrated[14], the F pocket can be occupied by GM with methionine as anchor residue. Methionine in the *C*-terminal position is not preferred but only tolerated by HLA-A\*02:01, as reported before[51]. Compared to MV5, which can bind with either valine, the second pentapeptide, NM5, therefore does not show the same strong binding behavior in our experiments. Parker et al.[49] claimed that GLGGGGGGV, carrying the minimal binding motif, is not sufficient for the stabilization of HLA-A\*02:01. However, GV9 binds well in our nMS experiment, and GV9 also increases thermal stability. Nevertheless, the weaker binding compared to NV9 shows that other amino acids contribute substantially to the binding energy.

From the shorter peptide binding data, we propose that the disulfide-stabilized MHC class I molecule has at least two positions that can be stabilized independently upon peptide binding (Fig. 8).

Our findings with modified peptide termini support published observations[7] that the single-modified nonapeptides can indeed stabilize MHC class I, albeit to a lesser extent. In contrast, the weak binding of the double-modified Ac-NV9-NH$_2$ confirms that one of the termini needs to be intact to form strong binding to either of the outer pockets. Thus, for example, undecapeptides with an overhang on both sides will have reduced affinity, as neither of the termini is intact. According to our data, the short, truncated peptides, which are unable to bind using both termini due to spatial limitations, can also stabilize their respective binding pocket individually, once again supporting our hypothesis concerning the independent stabilization of A and F pocket.

Binding of lipophilic small molecules into the class I binding groove has been shown several times, sometimes quite tightly and (because of alteration of the bound peptidome) with medical consequences[52]. In our MHC class I samples, erucamide is identified as the most prevalent contaminant, most likely from plastic ware[26]. It appears to bind specifically into the peptide-binding groove, since it is displaced by peptides. An alternative explanation, binding outside the groove and displacement by a peptide-induced conformational change, we consider less likely due to the structural similarity of empty and peptide-bound dsA2[14]. Thus, erucamide is used as a reference species to estimate the $K_d$ of the peptides: an increase of dsA2/pep with a simultaneous decrease of dsA2/erucamide demonstrates that the corresponding peptide actually occupies the space in the peptide pocket due to its affinity to dsA2 instead of just clustering nonspecifically on the protein. Interestingly, other lipids have been shown to bind class I[53,54], and since erucamide and related compounds are present in nutritional plants[55] and produced by *E. coli*[56], the binding could have biological relevance worth investigating.

HLA-A\*02:01 is one of the most prevalent A2 allotypes among Caucasians and Asians, and it has been shown that high binding affinity to A\*02:01 correlates with binding affinity to the entire A2 supertype, making A\*02:01 a suitable candidate for comprehensive peptide screening[51]. Epitope-based peptide vaccines offer various advantages regarding production, stability, and mutation risk. However, since HLA exhibits a strong polymorphism, the search for relevant, allele-spanning peptides has so far been challenging[51]. With nMS, we have wide latitude in our approach to identify and analyze potential epitopes. Measurements at low acceleration voltages yield good estimates for the peptides' binding affinities. In contrast, prediction tools such as NetMHC cannot predict the $K_d$ for shorter or modified peptides. Furthermore, we note inconsistencies between our data and the affinities predicted by NetMHC. While this is reasonable in terms of the numerical values, the relative affinities for our fairly large number of peptides are also incongruent with the predictions (Table S3). Moreover, the affinities determined by nMS and iDSF are in the same order of magnitude and correlate well. Thus, our work shows that the accuracy of affinity ranking by artificial neural networks needs to be debated.

Our measurements at higher energies yield a simple test for strong peptide binding. We show that pMHC gas-phase stability is mainly determined by side-chain interactions within the B and F pockets as well as by binding of the peptide termini. At an acceleration voltage of 50 V, only strongly bound ligands are retained. This value hence serves as a cut-off value in our nMS approach that could easily be employed in an nMS-based screening. Apart from some attempts[57–59], there is currently no high-throughput method for the identification of MHC class I binding peptides with immunogenic potential that is elaborated and reliable. Still, such high-throughput screenings are the ultimate key to the development of synthetic peptide vaccines that offer decisive advantages over conventional vaccines, as there is a lesser risk of unwanted host responses, and no possibility of reversion to pathogenic phenotypes, and no limitation for target diseases[1,60,61]. Vaccine production is rather easy and detached

from the natural source itself, which may be challenging to culture[1,61]. The greatest advantage of peptide vaccines lies in their stability. Especially in the context of the SARS-CoV-2 pandemic, mRNA vaccines have become very popular. However, especially these and other conventional vaccines are highly dependent on a continuous cold chain, whereas peptides show long-term thermal stability[1,60,62]. This feature is of particular importance, as the need for vaccination is often especially high in tropical and hot climates with limited medical infrastructure.

Taken together, our work provides a valid, sensitive, and rapid method to determine the $K_d$ of the MHC class I-peptide complex and also the basis to develop a novel high-throughput peptide screen for MHC class I epitopes. Since our technique is based on mass spectrometry, it allows working with very low sample consumption, and it also offers the possibility of simultaneous multi-species analysis.

## Methods

**Production of dsA2 molecules**. Production followed Anjanappa, Garcia-Alai et al.[14]. HLA-A*02:01(Y84C/A139C) disulfide mutant (dsA2) heavy chain and human β2m light chain were expressed in *Escherichia coli* using a pET3a plasmid. The proteins were extracted from inclusion bodies. The dsA2 complex was refolded in presence of 10 mM GM (*Bachem*), concentrated, and purified by size-exclusion chromatography on an ÄKTA system (*Cytiva*) using a *HiLoad 26/600 Superdex 200 pg column* (*Cytiva*).

**Native mass spectrometry**. Prior to native MS measurements, *Micro Bio-Spin 6 Columns* (molecular weight cutoff 6 kDa; *Bio-Rad*) were used at 1000 × g and 4 °C to exchange purified protein samples to 250 mM ammonium acetate (99.99% purity; *Sigma-Aldrich*), pH 8.0 as buffer surrogate. For native MS experiments, the final concentration of the dsA2 protein was 10 µM. Total peptide concentration (*Genecust*) ranged between 50 and 200 µM. Protein and peptides were incubated for at least 10 min on ice. No peptidic contaminants either in the bound state or in the free form in the low *m/z* region were detected. Native MS analysis was implemented on a *Q-Tof II* mass spectrometer (*Waters/Micromass*) in positive electrospray ionization mode. The instrument was modified to enable high mass experiments (*MS Vision*)[63]. Sample ions were introduced into the vacuum using homemade capillaries via a nano-electrospray ionization source (source pressure: 10 mbar). Borosilicate glass tubes (inner diameter: 0.68 mm, outer diameter: 1.2 mm; *World Precision Instruments*) were pulled into closed capillaries in a two-step program using a squared box filament (2.5 mm × 2.5 mm) within a micropipette puller (*P-1000, Sutter Instruments*). The capillaries were then gold-coated using a sputter coater (5.0 × 10⁻² mbar, 30.0 mA, 100 s, three runs to vacuum limit $3.0 \times 10^{-2}$ mbar argon, the distance of plate holder: 5 cm; *CCU-010, safematic*). Capillaries were opened directly on the sample cone of the mass spectrometer. In regular MS mode, spectra were recorded at a capillary voltage of 1.45 kV and a cone voltage of 150 V. Protein species with the quaternary structure were assigned by MS/MS analysis. These experiments were carried out using argon as collision gas ($1.2 \times 10^{-2}$ mbar). The acceleration voltage ranged from 10 to 100 V. Comparability of results was ensured as MS quadrupole profiles and pusher settings were kept constant in all measurements. The instrument settings of the mass spectrometer were optimized for non-denaturing conditions. A spectrum of cesium iodide (25 g/L) was recorded on the same day of the particular measurement to calibrate the data.

All spectra were evaluated regarding experimental mass (*MassLynx V4.1, Waters*), full width at half maximum (FWHM; *mMass*, Martin Strohalm[64]) and area under the curve (AUC; *UniDec*, Michael T. Marty[65]) of the detected mass species. The values of the shown averaged masses, FWHM (Table S1) and AUC (Table S2) of the different species as well as the corresponding standard deviation result from at least three independent measurements. Narrow peak widths indicate rather homogeneous samples. In order to eliminate nonspecific ESI clustering within the results, the raw data were corrected using the dsA2/pep fraction of the negative control YF9. Affinity $K_{d,high}$ was calculated directly from AUC measured at 150 V cone voltage and 10 V acceleration voltage, whereas $K_{d,low}$ was indirectly derived from the AUC of the dsA2/erucamide fraction. Since the cone voltage is linearly proportional to the ISD, the distribution of peptide-bound and peptide-unbound dsA2 at non-dissociating conditions can be estimated. According to the equation ($occupancy = -0.0052 \times cone\ voltage + 1.2476$) derived from the linear relationship of cone voltage and occupancy, the binding groove is fully occupied at 48 V and thus reflects the in-solution conditions of the superstoichiometric mixture. The peptide-free (erucamide-bound) fraction of the protein measured at a cone voltage of 150 V can be corrected using the linear equation. Subsequently, if this fraction is subtracted from the possible 100% occupancy, the fraction that binds the peptide is obtained.

**Differential scanning fluorimetry**. Thermal stability and binding affinity were determined using nanoscale differential scanning fluorimetry on a *Prometheus*

NT.48 (*NanoTemper Technologies*). Capillaries were filled with 10 µL of respective samples in duplicates and loaded into the reading chamber. The scan rate was 1 °C/min ranging from 20 to 80 °C for thermal stability and 95 °C for binding affinity measurements. Protein unfolding was measured by detecting the temperature-dependent change in intrinsic tryptophan fluorescence at emission wavelengths of 330 and 350 nm.

**Thermal stability (nDSF)**. About 2 µM of empty dsA2 were dissolved in citrate phosphate buffer, pH 7.6[66], and incubated with peptidic ligands (0.2 µM to 2 mM) on ice for 30 min Melting curves and $T_m$ values were generated by *PR.ThermControl V2.1* software (*NanoTemper Technologies*) using the first derivative of the fluorescence at 330 nm.

**Binding affinity (iDSF)**. Empty dsA2 dissolved in 20 mM Tris pH 8, 150 mM NaCl was incubated with peptidic ligands at different concentrations depending on their predicted or assumed $K_d$ range. For each peptide, a twofold serial dilution series (11 concentrations) was prepared, while protein concentration was kept constant at 2.2 µM. A pure protein was analyzed as well. *PR.ThermControl V2.1.2* software (*NanoTemper Technologies*) was used to control the device. Data processing and evaluation was executed via the *FoldAffinity* web server (*EMBL Hamburg*, https://spc.embl-hamburg.de/[39]).

**Statistics and reproducibility**. Concerning the native mass spectrometry data, the values of the shown averaged masses, FWHM, and AUC of the different species as well as the corresponding standard deviation result from at least three independent measurements. When feasible (Fig. 3), all individual measured points are shown in addition to the mean value and the error bars representing the standard deviation. For the presentation of indirectly determining quantities such as the $K_d$, the standard deviation was obtained according to the rules of Gaussian error propagation. Differential scanning fluorimetry was performed in duplicates. The data were expressed as the mean ± standard deviation of the mean.

**Reporting Summary**. Further information on research design is available in the Nature Research Reporting Summary linked to this article.

## Data availability

The mass spectrometry proteomics data have been deposited to the ProteomeXchange Consortium via the PRIDE[67] partner repository with the dataset identifier PXD027725. Other data that support the findings of this study are available from the corresponding author upon reasonable request.

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

## Acknowledgements

The Leibniz Institute of Virology (LIV) is supported by the Freie und Hansestadt Hamburg and the Bundesministerium für Gesundheit (BMG). J.-D.K. and C.U. acknowledge support by the Leibniz ScienceCampus InterACt. We thank Thomas Dülcks, University of Bremen, for identifying the erucamide in our samples. We acknowledge technical support from the SPC facility at EMBL Hamburg. A.S. and S.S. thank Uschi Wellbrock for excellent technical assistance. C.U. acknowledges the Leibniz Association grant SAW-2014-HPI-4 and S.S. the Deutsche Forschungsgemeinschaft grant SP583/12-1. Molecular graphics images were produced using the UCSF Chimera package from the Resource for Biocomputing, Visualization, and Informatics at the University of California, San Francisco (supported by NIH P41 RR-01081).

## Author contributions

The manuscript was written with the contributions of all authors. It was conceptualized by C.U., J.-D.K., and S.S. with methodological advice given by M.G.-A. C.U. supervised the project. Experiments were performed by A.S., C.G., and J.-D.K. Data were analyzed by J.-D.K. and S.N., partially using software provided by S.N. Data visualization was conducted by J.-D.K. The original draft was written by J.-D.K. Review and editing were done by C.U., J.-D.K., and S.S. with the input of all other authors. C.U., S.S., and M.G.-A. acquired funding for the project.

## Funding

## Competing interests

The authors declare no competing interests.
