## [Peer Review File · Communications Biology]

Reviewers' comments:

Reviewer #1 (Remarks to the Author):

Kopicki et al. have done a good job in testing the bonding of several peptides to disulfide stabilized HLA-A*02:01 molecule. They used native mass spectrometry to determine the K_d of the complexes loaded with the peptides. In the series of sixteen examined peptides results showed that K_d correlates well with thermal stabilization, and in the case of modified charged peptides effect of double occupancy was shown. This information could be of interest to the researchers synthesizing peptides and investigating their binding to the MHC molecules in some particular cases, i.e. vaccine research.

Authors have well-explained effects occurring upon peptide binding, and they also mention that binding of erucamide to dsA2 molecule as a contaminant from plasticware was observed. What would be other routes to confirm this (i.e. isotopically labeled molecule), or alternatively is it possible to use a different type of laboratory consumables, i.e. glass where this binding does not occur?

Also, here are a few comments to clarify on some points:

Pg 2 line 42: "...becomes undetectable by nMS.."

It is suggested to modify this sentence to indicate that peptide wasn't detected after SEC rather than being undetectable.

Pg 3 line 60: What was incubation time?

Pg 6 Fig 4: Figure is difficult to read due to low quality, please provide better resolution Figure 4.

Reviewer #2 (Remarks to the Author):

This manuscript described a promising method to measure the binding affinity of HLA protein with antigens that presented by the class I MHC system, which is important to understand our adaptive immune system and design vaccines against infectious, autoimmune disease and cancer. The authors proved this accuracy and reliability of this method with pre-established model peptides and investigated several factors that would affect the performance of the method. It's a well-designed approach and solid study. I am looking forward to seeing expanded applications of this method on how mutations in cancer neoantigens and viral antigen proteins affected the binding affinity to different HLA alleles.

Minor issues

1 line 30-32 should be elaborated more with specific examples to demonstrate the dissociation can be measured in gas phase.

2 Dissociation of beta-2-microglobulin was observed (as shown in Figure 1 C, D). It raises question about the how the dissociation of the whole MHC in gas phase. Is the dissociation of peptide the first step or beta-2-microglobulin? If beta-2-microglobulin dissociates first, how to estimate the factor and correct its impact on the calculation of constant?

3 Typo in line 71, 44.071 should be 44,071

4 In Figure 2, a percentage based AUC was used to measure the relative abundance of difference species. How was the peak area normalized? Did empty dsA2 and dsA2/peptide complex have similar ionization efficiency?

5 Figure 3, it' s confusing that there are two linear fitted curves but only one equation is provided. Similar curves for dsA2/peptide should also be provided.

6 Resolution for Figure 4 need to be improved

7 The authors proved that neutralizing the terminal charges of the peptides would reduce the binding affinity. However, both N-terminal acetylation and C-terminal amidation are rare to see in natural MHC associated peptides. It would be more interesting to see if PTMs on the side chains (phosphorylation, methylation etc.) change the binding affinity.

Reviewer #3 (Remarks to the Author):

In this paper, Kopicki et al. report a novel approach to assess the binding affinity of peptide epitopes to MHC Class I molecules based on native mass spectrometry. The authors make use of their previously reported disulfide-stabilized recombinant HLA-A*02:01 molecule (dsA2) which they load with peptide epitopes of interest. Native mass measurement at low acceleration speed then allows to measure the fractional amounts of free dsA2, dsA2 loaded with peptide epitope and dsA2 loaded with erucamide – a plastic derived contaminant – as most important complexes. In an elegant way, the authors use erucamide as a reference species for peptide binding measurements, taking advantage of the linear relationship of cone voltage and occupancy of dsA2 with erucamide due to in-source dissociation (ISD). Using the NV9 epitope from human cytomegalovirus pp65 and variants thereof, they show good correlation of the measured K_d values with iDSF and nDSF data. Finally, the authors apply their approach to evaluate the impact of NV9 truncations, neutralizing terminal charges and synergistic binding of two short peptides, measurements that suggest that dsA2 can stabilize its A and F pockets independently, in line with previous results.

The manuscript is well written and properly constructed, and the experimental data is of high quality. The NV9 data validates the procedure which indeed holds great potential to screen for high affinity epitopes, important for (peptide) vaccine development. Hence, I am generally in support of publication of this study, but I do have a few minor comments that need to be addressed:

1) While it seems a good idea (and necessary) to use erucamide as a reference species for K_d measurements, I wonder whether the dependency on such (non-controlled) contaminant poses certain risks to the method? Do certain sample preparation steps strictly require plastic ware? Would it not be safer to add exogenous erucamide to the samples in a controlled way? At least the authors should comment on this potential limitation in the discussion.

2) line 102: ...the obtained K_d for NV9 is only 8±2 μM... please change to ...the obtained K_d (K_{d,high} defined below) for NV9 is only 8±2 μM...

3) Please include a figure citation for Fig. 2B and carefully check all other figure and table citations throughout the text.

4) line 184: ... which no longer carries terminal charges...

Rebuttal letter

Reviewer #1 (Remarks to the Author):

Kopicki *et al.* have done a good job in testing the bonding of several peptides to disulfide stabilized HLA-A*02:01 molecule. They used native mass spectrometry to determine the K_d of the complexes loaded with the peptides. In the series of sixteen examined peptides results showed that K_d correlates well with thermal stabilization, and in the case of modified charged peptides effect of double occupancy was shown. This information could be of interest to the researchers synthesizing peptides and investigating their binding to the MHC molecules in some particular cases, i.e. vaccine research.

Authors have well-explained effects occurring upon peptide binding, and they also mention that binding of erucamide to dsA2 molecule as a contaminant from plasticware was observed. What would be other routes to confirm this (i.e. isotopically labeled molecule), or alternatively is it possible to use a different type of laboratory consumables, i.e. glass where this binding does not occur?

We thank the reviewer for this important suggestion. To the best of our knowledge, we have used conventional and native mass spectrometry to identify the ligand as erucamide. However, other detection methods are tedious to employ and optimize (see below). The exact identity of the compound is not crucial as long as it is reproducibly present. To account for this slight ambiguity, we rephrased the text stating it is "most likely ... erucamide." (p. 2, l. 55)

We would like to provide some additional information (see also reply to reviewer #3). Even if there was a complete change to glassware in protein production, there would still be an issue with buffer exchange. All common devices that can be used for buffer exchange (spin filters, dialysis membranes, spin columns) are at least partly made of plastic. We contacted the manufacturers Bio-Rad (Micro Bio-Spin™ P-6 Gel Columns) and Sartorius (Vivaspin 500, 30,000 MWCO) to get more information about the composition of the plastic and they cannot exclude erucamide in their products.

In previous experiments, we had already attempted to introduce external erucamide (both organically dissolved and as a solid directly in the protein solution) into the sample. This did not result in any changes compared with the "pure" protein sample. According to our calculations, the protein in solution is fully occupied by erucamide. This is consistent between protein batches (see now p. 2, l. 57). An exchange only occurs when affine peptides displace the erucamide. Accordingly, it is difficult to detect this contaminant via other routes. Interestingly, we also found that protein from a batch of poor quality that was no longer able to bind peptide had also failed to bind erucamide. Apparently, the presence of erucamide is also an indication that the protein can still function natively.

Also, here are a few comments to clarify on some points:

Page 2, line 42: “..becomes undetectable by nMS..”

It is suggested to modify this sentence to indicate that peptide wasn't detected after SEC rather than being undetectable.

Thanks for the suggestion, has been changed.

Page 3, line 60: What was incubation time?

The incubation time of protein and peptide was always at least ten minutes before the mass spectrometry measurements. The mixtures were also kept on ice for longer than

ten minutes, but this did not lead to detectable deviations. The incubation time is now given in the method section (p. 13 l.374).

Page 6, Fig 4: Figure is difficult to read due to low quality, please provide better resolution Figure 4.

For unknown reasons all figures were of low quality in the submission manuscript and have been replaced by higher resolution versions.

Reviewer #2 (Remarks to the Author):

This manuscript described a promising method to measure the binding affinity of HLA protein with antigens that presented by the class I MHC system, which is important to understand our adaptive immune system and design vaccines against infectious, autoimmune disease and cancer. The authors proved this accuracy and reliability of this method with pre-established model peptides and investigated several factors that would affect the performance of the method. It's a well-designed approach and solid study. I am looking forward to seeing expanded applications of this method on how mutations in cancer neoantigens and viral antigen proteins affected the binding affinity to different HLA alleles.

Minor issues

1 line 30-32 should be elaborated more with specific examples to demonstrate the dissociation can be measured in gas phase.

The determination of dissociation constants by native mass spectrometry is a method that was established years ago. We agree that the common reader may not be aware of this and have added a few specific examples (p. 2, l. 32-39):

“Moreover, a small ligand has only minor influence on the ionization efficiency of a much larger protein complex. For example, Garcia-Alai & Heidemann et al.¹ have determined the dissociation constants for clathrin-associated adapter protein-phospholipid complexes. The work of Jecklin et al.² also demonstrates how nMS can be used to quantitatively assess binding affinity using multiple protein-ligand systems (hen egg white lysozyme and *N,N',N''*-triacetylchitotriose; adenylate kinase and adenosine-5'-diphosphate; lymphocyte-specific kinase and a respective inhibitor). Fundamental in all examples is that the peak area of the individual mass species is integrated and then the concentrations of protein, ligand and the resulting complex are calculated.”

2 Dissociation of beta-2-microglobulin was observed (as shown in Figure 1 C, D). It raises question about the how the dissociation of the whole MHC in gas phase. Is the dissociation of peptide the first step or beta-2-microglobulin? If beta-2-microglobulin dissociates first, how to estimate the factor and correct its impact on the calculation of constant?

These are important points that nevertheless do not hamper our analysis. First of all, only in absence of a bound peptide (see figure 1 and 2) a significant amount of free β_2m is detected. Importantly, in the presence of peptide and at the measurement conditions used to determine the K_d , there is a negligible amount of free β_2m and heavy chain (hc) being no larger than the measurement error. Both dissociation pathways (dsA2+pep vs. β_2m +hc) occur independently of the peptide of interest (see now p. 2 l. 52-53) and simultaneously, thus, are in competition. In absence of peptide, β_2m and hc dissociation

¹ Garcia-Alai u. a., „Epsin and Sla2 form assemblies through phospholipid interfaces“.

² Jecklin u. a., „Which electrospray-based ionization method best reflects protein-ligand interactions found in solution? a comparison of ESI, nanoESI, and ESSI for the determination of dissociation constants with mass spectrometry“.

is the sole and hence populated pathway. In presence of peptide, the protein complex is stabilized and the dsA2 peptide dissociation becomes dominant. The affinity and identity of the peptide further influences the pathway preference. In the case of the high-affinity NV9, for example, it can be seen quite clearly that half of the free hc is still bound to the NV9 (see Fig. 2B). This is not the case for low-affinity peptides.

This nicely shows the beauty of the erucamide approach. Although β_2m and hc dissociate, we can perform an accurate affinity determination ($K_{d,low}$) as the erucamide *always* dissociates first, regardless of the cone voltage or the peptide affinity. The remaining portion of the protein, which is occupied by erucamide, therefore indicates how much has been displaced by the peptide.

3 Typo in line 71, 44.071 should be 44,071

Done.

4 In Figure 2, a percentage based AUC was used to measure the relative abundance of difference species. How was the peak area normalized? Did empty dsA2 and dsA2/peptide complex have similar ionization efficiency?

The peak area of dsA2, dsA2/pep, dsA2/pep/pep and dsA2/erucamide was summed. From this 100%, the respective proportion of the different species was presented. Beyond that, there was no normalization (see methods).

The nonapeptides are still very small relative to the protein complex. Small ligands of this type have only a marginal effect on the ionization efficiency, which is now clearly stated in l. 32-33. In our approach based on the erucamide, the peptide affinity is decoupled from the ionization efficiency of the dsA2 peptide complex being analyzed. Hereby, only the erucamide's influence on ionization efficiency is important, and this is the same for all experiments and even smaller than for peptides.

5 Figure 3, it' s confusing that there are two linear fitted curves but only one equation is provided. Similar curves for dsA2/peptide should also be provided.

Both linear fitted curves are derived from the same dataset and differ in the slope only by the sign. For clarity, we have now inserted both formulas and color-coded the formulas in accordance with the curves.

In Fig. 3B, it can be seen how the signals of the NV9-dsA2 complex are affected by the cone voltage. Using the K_d determination via the erucamide-bound protein fraction, we exploit the fact that the cone voltage's impact is always the same, regardless of which peptide is analyzed. In our opinion, the display of the cone voltage dependence by an equation for a single peptide would not provide additional value as it is peptide specific. By using the erucamide, we avoid the need to determine the exact linear dependence for each analyte individually in advance. We added the following explanation for Fig. 3B: "The proportion of the dsA2/pep species is likewise linearly dependent on the cone voltage (Fig. 3B). The extent to which the protein-peptide complex is affected by ISD depends on the gas phase properties and the binding affinity of the peptide in question."(p.4 l. 115-118).

6 Resolution for Figure 4 need to be improved

As mentioned above, we have changed the graphic formats so that all images are now available in adequate quality. Thanks for pointing this out!

7 The authors proved that neutralizing the terminal charges of the peptides would reduce the binding affinity. However, both N-terminal acetylation and C-terminal amidation are rare to see in natural MHC associated peptides. It would

be more interesting to see if PTMs on the side chains (phosphorylation, methylation etc.,) change the binding affinity.

We performed the charge deletion on NV9 to test the termini's role in binding. In addition, this approach simulates the binding of overhanging decapeptides and undecapeptides, which naturally lack charge at these positions (see p. 12, l. 321; now also p. 6, l. 168-169). Moreover, N-terminal acetylations on peptides are by no means uncommon *in vivo*^{3,4}, which is now also stated alongside on p.6. It is a very nice idea, for future studies, to analyze peptides with other modifications. Hereby, it would probably be most interesting to see how modifications at anchor positions affect binding.

Reviewer #3 (Remarks to the Author):

In this paper, Kopicki *et al.* report a novel approach to assess the binding affinity of peptide epitopes to MHC Class I molecules based on native mass spectrometry. The authors make use of their previously reported disulfide-stabilized recombinant HLA-A*02:01 molecule (dsA2) which they load with peptide epitopes of interest. Native mass measurement at low acceleration speed then allows to measure the fractional amounts of free dsA2, dsA2 loaded with peptide epitope and dsA2 loaded with erucamide – a plastic derived contaminant – as most important complexes. In an elegant way, the authors use erucamide as a reference species for peptide binding measurements, taking advantage of the linear relationship of cone voltage and occupancy of dsA2 with erucamide due to in-source dissociation (ISD). Using the NV9 epitope from human cytomegalovirus pp65 and variants thereof, they show good correlation of the measured K_d values with iDSF and nDSF data. Finally, the authors apply their approach to evaluate the impact of NV9 truncations, neutralizing terminal charges and synergistic binding of two short peptides, measurements that suggest that dsA2 can stabilize its A and F pockets independently, in line with previous results.

The manuscript is well written and properly constructed, and the experimental data is of high quality. The NV9 data validates the procedure which indeed holds great potential to screen for high affinity epitopes, important for (peptide) vaccine development. Hence, I am generally in support of publication of this study, but I do have a few minor comments that need to be addressed:

1) While it seems a good idea (and necessary) to use erucamide as a reference species for K_d measurements, I wonder whether the dependency on such (non-controlled) contaminant poses certain risks to the method? Do certain sample preparation steps strictly require plastic ware? Would it not be safer to add exogenous erucamide to the samples in a controlled way? At least the authors should comment on this potential limitation in the discussion.

We raised similar question, when we identified erucamide, although coming from a different angle and so did reviewer #1. In several attempts, we tried to remove or supplement erucamide but never observed significant differences. It turned out that the dsA2 in all preparations (observed over several years) was already saturated and only upon addition of high affinity peptides less erucamide occurred. We conclude that contact of the smallest volume of protein solution with the macroscopic plastic is sufficient for complete occupation of dsA2 with erucamide. This perception triggered our indirect erucamide approach. Importantly, it is currently impossible to transfer the protein into

³ Arnesen, „Towards a Functional Understanding of Protein N-Terminal Acetylation“.

⁴ Gautschi u. a., „The Yeast N(Alpha)-Acetyltransferase NatA Is Quantitatively Anchored to the Ribosome and Interacts with Nascent Polypeptides.“

nMS compatible solutions without using potentially contaminated plastic ware – we now even enquired with suppliers.

The erucamide occupies all dsA2 molecules as long as it is not displaced by the peptide (see also reply to reviewer #1). How much erucamide is initially displaced in solution depends on the affinity of the peptide, which we can thus calculate. Of course, since we have not yet tracked down the actual origin of erucamide, we cannot predict when it will appear and when it will not. The addition of exogenous erucamide has been tried (see above) but has not proved to be useful. We clarify that this is a consistent and not single batch observation now in the main text section (p. 2, l. 57).

We have shown that we can calculate peptide affinities in the presence of the erucamide, but we would like to emphasize that it is of course possible to determine K_d s by native MS in the absence of such a reference species, as we have already explained in the manuscript. Before the discovery of the erucamide in our samples, we had of course also expected to follow the conventional route via the dsA2 and dsA2/pep concentration alone, as described for example in Garcia-Alai & Heidemann *et al.*⁵ and Jecklin *et al.*⁶ (see p. 4, l. 32-39)

2) line 102: ...the obtained K_d for NV9 is only $8 \pm 2 \mu\text{M}$... please change to ...the obtained K_d ($K_{d,\text{high}}$ defined below) for NV9 is only $8 \pm 2 \mu\text{M}$...

Done.

3) Please include a figure citation for Fig. 2B and carefully check all other figure and table citations throughout the text.

Done.

4) line 184: ... which no longer carries terminal charges...

Done.

⁵ Garcia-Alai u. a., „Epsin and Sla2 form assemblies through phospholipid interfaces“.

⁶ Jecklin u. a., „Which electrospray-based ionization method best reflects protein-ligand interactions found in solution? a comparison of ESI, nanoESI, and ESSI for the determination of dissociation constants with mass spectrometry“.

REVIEWERS' COMMENTS:

Reviewer #1 (Remarks to the Author):

Dear authors,

Thank you for revising your manuscript and taking into account the reviewer's concerns.

Reviewer #2 (Remarks to the Author):

The authors have addressed all the concerns properly.

Reviewer #3 (Remarks to the Author):

I appreciate the efforts of the authors to address my comments. The manuscript can now be accepted for publication.

There are still a few errors in literature citations (e.g. page 2 line 54), but I assume these will be resolved during typesetting of the manuscript.

Reviewer #1 (Remarks to the Author):

Dear authors,

Thank you for revising your manuscript and taking into account the reviewer's concerns.

Reviewer #2 (Remarks to the Author):

The authors have addressed all the concerns properly.

Reviewer #3 (Remarks to the Author):

I appreciate the efforts of the authors to address my comments. The manuscript can now be accepted for publication.

There are still a few errors in literature citations (e.g. page 2 line 54), but I assume these will be resolved during typesetting of the manuscript.

We thank the reviewers for their time and dedication. We have carefully checked all references, which should now be correct.